# Oxidative Stress—A Key Player in the Course of Alcohol-Related Liver Disease

**DOI:** 10.3390/jcm10143011

**Published:** 2021-07-06

**Authors:** Agata Michalak, Tomasz Lach, Halina Cichoż-Lach

**Affiliations:** 1Department of Gastroenterology with Endoscopy Unit, Medical University of Lublin, Jaczewskiego 8, 20-090 Lublin, Poland; agatamichalak@umlub.pl; 2Department of Orthopedics and Traumatology, Medical University of Lublin, Jaczewskiego 8, 20-090 Lublin, Poland; tomaszlach@umlub.pl

**Keywords:** oxidative stress, alcohol-related liver disease, micro-RNA, sirtuin gene family

## Abstract

Oxidative stress is known to be an inseparable factor involved in the presentation of liver disorders. Free radicals interfere with DNA, proteins, and lipids, which are crucial in liver metabolism, changing their expression and biological functions. Additionally, oxidative stress modifies the function of micro-RNAs, impairing the metabolism of hepatocytes. Free radicals have also been proven to influence the function of certain transcriptional factors and to alter the cell cycle. The pathological appearance of alcohol-related liver disease (ALD) constitutes an ideal example of harmful effects due to the redox state. Finally, ethanol-induced toxicity and overproduction of free radicals provoke irreversible changes within liver parenchyma. Understanding the underlying mechanisms associated with the redox state in the course of ALD creates new possibilities of treatment for patients. The future of hepatology may become directly dependent on the effective action against reactive oxygen species. This review summarizes current data on the redox state in the natural history of ALD, highlighting the newest reports on this topic.

## 1. Introduction

Oxidative stress is a crucial factor responsible for the pathological appearance of diverse systemic entities. Neoplasmatic process, cardiovascular diseases, aging, and many more phenomena are inseparably related to reactive oxygen species (ROS). Acute and chronic liver disorders also present a well-known target for free radical activity. Oxidative stress constitutes a major triggering factor in the course of alcohol-related liver disease (ALD) [1,2,3,4]. Alcohol-induced liver disorders involve a broad range of molecular injuries of hepatocytes, including steatosis, steatohepatitis, development of cirrhosis, and a possible transformation to hepatocellular carcinoma (HCC) [5,6,7]. According to worldwide data, alcohol-related liver cirrhosis (ALC) is a third cause of alcohol-derived deaths [8,9]. This review highlights the impact of oxidative stress on the progression and complications of ALD, summarizing already collected data. Approximately 2–10% of absorbed ethanol is eliminated via the lungs and kidneys; the major residue is metabolized mostly by oxidative pathways in the liver and due to nonoxidative mechanisms in the extrahepatic tissues [10,11,12,13]. The liver injury due to alcoholic toxicity comprises a broad range of pathologies. First, DNA, proteins, and lipids are prone to be damaged by a crucial metabolite of ethanol—acetaldehyde—together with other highly reactive oxidants [14,15]. Altered hepatic respiration and lipid metabolism are followed by hypoxia and impaired mitochondrial function [16,17,18]. Moreover, acetaldehyde-protein adducts alter signaling pathways and ion channel function [19,20]. As the result, hepatocytes die and provoke further mediation of pro-inflammatory particles, leading to tissue repair and gradual fibrogenesis within the liver, mediating the development of hepatocellular carcinoma [21,22,23,24]. Simultaneously, alcohol-induced autophagy followed by apoptotic cell death appears to be another crucial mechanism in hepatocellular injury [25,26,27,28,29]. Figure 1 shows a natural history of ALD. 

## 2. Alcohol, High Fat Diet and Mitochondria

### 2.1. Ethanol Metabolism and Oxidative Stress

A major pathway of alcohol metabolism in the liver is an oxidative one, which leads through its metabolism to acetaldehyde by alcohol dehydrogenases, cytochrome P4502E1 (CYP2E1), and catalase [30,31,32]. Because of a broad spectrum of enzymes capable of alcohol metabolism, this process takes place in various tissues, however, the liver is the primary organ. The second minor, non-oxidant pathway of alcohol breakdown is regulated via fatty acid ethyl ester synthase and phospholipase D with the formation of fatty acid ethyl ester and phosphatidyl ethanol [33]. CYP2E1 belongs to the P450 enzyme family that has a key role in alcohol, drug, toxin, lipid, and carcinogen metabolism. In human organisms CYP2E1 is mainly expressed in hepatocytes. Its function is to metabolize substrates into more polar particles—for secretion or conversion by other microsomal phase II enzymes [34]. CYP2E1 also transfers active electrons from reduced nicotinamide adenine dinucleotide phosphate (NADPH) or reduced nicotinamide adenine dinucleotide (NAD) to oxygen and leads to the production of ROS with this mechanism. Toxic metabolites derived from CYP2E1 activity, together with coexisting oxidative stress, are well known triggering factors responsible for liver injury by exacerbating an inflammatory and fibrogenic response, reflected by recruitment of leukocytes and hepatic stellate cells (HSCs) [35,36]. Chronic alcohol consumption was proved to increase the expression of CYP2E1 protein [37,38]. On the other hand, knocking out CYP2E1 in previous models reduced alcohol-induced hepatic oxidative stress and prevented the development of alcoholic steatosis [39,40]. ALD might promote mitochondrial destruction and dysfunction due to excessive oxidative stress. Physiologically, approximately 1–2% of oxygen leaks out as ROS from the mitochondrial electron transport chain (ETC). These active molecules are essential for the regulation of various cellular signaling loops and their excess is neutralized by cellular antioxidant complexes with no harm for the cell [41,42,43]. Nevertheless, under pathological conditions, with coexisting exposure to certain toxic agents (e.g., alcohol, high fat diets), the release of ROS from mitochondrial ETC becomes too high. Interestingly, cholesterol overload might diminish the expression of key DNA repair genes, exacerbating oxidative damage to the liver and even promoting the development of liver cancer [44,45]. Furthermore, mitochondria, which are well known as a primary source of free radicals, become paradoxically the main target of oxidative damage because they contain relatively low levels of antioxidants, such as a reduced glutathione (GSH) [46,47,48,49]. Its concentration in the cytosol is definitely higher, because a special transporter protein is required to move GSH directly to mitochondria, where GSH is not synthesized. Of note, chronic alcohol exposure alters the function of the GSH transporter channel, resulting in a progressive deficiency of GSH within mitochondria [50,51]. Previous investigations conducted on mitochondria exposed to a high level of oxidative stress confirm this theory—mitochondria from those models present irregular shapes and altered functions. Of note, oxidative stress due to excessive alcohol ingestion can downregulate alcohol dehydrogenase activity, protecting the liver from further injury. Chronic intake of ethanol leads finally to the stage of metabolic adaptation (tolerance), in which an increased rate of blood ethanol clearance is observed. Another causative factor for this situation is believed to be that substrate shuttle capacity and transport of reducing equivalents into the mitochondria is not disturbed by chronic alcohol consumption. On the other hand, according to the hypermetabolic state hypothesis, changes in thyroid hormone levels increase (Na^+^ + K^+^)-activated ATPase, followed by elevated ADP concentration. This increases the state 3 mitochondrial oxygen consumption, intensifying NADH reoxidation. Increased oxygen consumption may become the reason for hypoxia, especially to hepatocytes of zone 3 of the liver acinus, the region where alcohol toxicity originates (centrilobular hypoxia hypothesis) [52,53,54].

### 2.2. Nitrosative Stress

Other cellular enzymes are also able to generate ROS and reactive nitrogen species (RNS) including nitric oxide (NO). This group of enzymes comprises myeloperoxidase and nicotinamide adenine dinucleotide phosphate hydrogen (NADPH) oxidase in phagocytic immune cells, ethanol-inducible CYP2E1 and cytochrome P4504A (CYP4A) isozymes in endoplasmic reticulum (ER), cytosolic xanthine oxidase, and nitric oxide synthase isozymes including its inducible form (iNOS) in activated Kupffer cells together with recruited neutrophils [55,56]. Except for oxidative stress, nitrative stress constitutes another important metabolic condition, resulting from the reaction of ROS with NO. Excessive amounts of free radicals might lead to the overproduction of a potentially toxic peroxynitrite (ONOO−) in the presence of NO [57]. Peroxynitrite is an agent which can result in the modification of diverse proteins while nitrated tyrosine residues serve the function of a stable marker for nitrative stress. Another essential source of oxidative stress is intestinal NO. Indeed, alcohol-induced overproduction of NO by inducible nitric oxide synthase (iNOS) alters barrier function. The prevention of alcohol-induced NO overproduction in previous rat models restored proper barrier integrity. A certain mechanism responsible for alcohol-induced gut leakiness has not been fully elucidated, however it appears that miRNA might be potentially involved in this cascade. ZO-1 (zonula occludens-1), which belongs to crucial tight junctional proteins implied in the regulation of intestinal barrier, is a target gene of miR-212. Colon biopsies obtained from ALD patients revealed its overexpression. Consequently, alcohol-induced higher concentration of intestinal miR-212 in cell cultures was observed together with downregulation of ZO-1. Additionally, alcohol-induced miR-212 overexpression and disruption of ZO-1 morphology in cell cultures were significantly inhibited when iNOS was knocked down. It was confirmed in iNOS knock-out (KO) mice model fed with alcohol, indicating a close dependency between NO and miRNAs. These findings support the idea that iNOS serves an important role in alcohol-induced miR-212 overexpression, which disrupts intestinal barrier integrity by inhibiting ZO-1 expression [58]. Interestingly, too high a level of ROS and RNS even suppresses the action of antioxidants (mitochondrial superoxide dismutase (SOD_2_), catalase, glutathione peroxidase and glutathione reductase, and vitamins). In such pathological circumstances, mitochondrial DNA may undergo oxidation, nitrosation, and/or nitration, which is finally reflected by mitochondrial dysfunction [59,60,61]. Of note, mitochondrial DNA (mtDNA) might undergo oxidation due to prolonged oxidative stress in alcoholics. Peroxynitrite derived from the spontaneous reaction of NO with superoxide leads to mtDNA depletion. Consistently, concentration of 8-hydroxy-2′-deoxyguanosine together with mutations and strand breaks of mtDNA increase. Finally, even multiple mtDNA deletions occur, proved in liver tissues from patients with ALD. Despite the presence of hundreds of copies of mtDNA, their abundant structural aberrations can be followed by attenuated mitochondrial respiration and ATP synthesis, aggravating hepatocyte injury. Structural disturbances of mtDNA may involve its D-loop region, responsible for the replication and maintenance of mtDNA [62]. As a consequence, expression of mtDNA replication-related proteins, (e.g., mitochondrial single-stranded DNA-binding protein, mitochondrial transcription factor A) diminishes [63,64]. Morphologically changed mtDNA also becomes the target for mitochondrial endonuclease G and for this reason the production of mt-DNA encoded key proteins of the oxidative phosphorylation system is impaired [65,66,67,68]. Table 1 presents various molecules participating in the different stages of the redox state due to ALD. 

### 2.3. Lipids, Steatosis and Steatohepatitis in ALD

Metabolism of ethanol in hepatocytes by CYP2E1 is inseparably connected with overproduction of ROS. Oxidative stress promotes lipid peroxidation, protein carbonylation, and formation of 1-hydroxyethyl radical and lipid radical formation. CYP2E1 also stimulates Ω-1-hydroxylation of endobiotic substrates, e.g., fatty acids, steroids, and prostaglandins. This conversion takes place in microsomes and constitutes an alternative for long chain fatty acid mitochondrial β-oxidation [69]. An essential role of Ω-1-hydroxylation of arachidonic acid (AA) has been emphasized due to a proinflammatory profile of AA-derived eicosanoids [70]. Excessive alcohol consumption facilitates the hydroxylation of AA and other polyunsaturated fatty acids (PUFAs) within microsomes [71,72]. Additionally, a lower level of AA in the liver was noticed in both murine and human ALD and supplementation of CYP2E1 inhibitor caused an increase in its concentration. Hepatocytes ballooning and lymphocyte infiltration characterize steatohepatitis—a prominent feature of ALD. The imbalance between de novo lipid synthesis and lipid β-oxidation accompanying alcohol consumption leads to the accumulation of lipid droplets in hepatic parenchyma [73]. Ethanol metabolism is also associated with up-regulation of sterol regulatory element binding protein 1c (SREBP-1c) and down-regulation of peroxisome proliferator activated receptor alpha (PPARα) [74,75]. Aberrated expression of the abovementioned receptors promotes fatty acid synthesis and simultaneously inhibits β-oxidation [76,77]. Mice fed with alcohol were found to develop more severe liver steatosis in comparison to pair-fed mice receiving the same caloric intake, showing that an alcohol-induced metabolic imbalance leads to steatosis [78]. Except for CYP2E1, cytosolic alcohol dehydrogenase and mitochondrial aldehyde dehydrogenase 2 are involved in the metabolism of ethanol; reducing equivalents (reduced NAD and NADPH) and acetyl-coenzyme A (CoA) equivalents (acetaldehyde and acetate) are created as the result. NADPH and acetate constitute the substrates of lipid β-oxidation but also participate in de novo lipogenesis. Thus, alcohol consumption interferes with lipid homeostasis and promotes the direction of lipogenesis to exacerbate alcoholic liver steatosis. Newly synthesized free fatty acids (FFAs) are transformed into diacylglycerol (DAG) and triacylglycerol (TAG) to create lipid droplets within hepatocytes [79,80]. Lipogenesis is limited by acetyl-CoA carboxylase (ACC) and its transcription is upregulated by factor Srebp-1. Uncontrolled lipid droplet accumulation and ROS are the reasons for hepatocyte ballooning and apoptosis [81]. Dead hepatocytes trigger inflammatory response within the liver, stimulating the release of proinflammatory agents (tumor necrosis factor alpha (TNF-α, interleukin (IL)-1b, IL-6, and transforming growth factor β1 (TGF-β1)). Moreover, neutrophils mediate the progression of destruction by intensifying oxidative stress and finally kill hepatocytes, creating a typical picture of alcoholic hepatitis [82,83]. Many previous investigations support the idea of a tight relation between ALD and oxidative stress. A decrease in the antioxidant enzyme glutathione peroxidase-1 during chronic alcohol consumption confirms this theory. Furthermore, collected data suggest that ROS cause damage among proteins, lipids, and cytoplasm [84]. Galicia-Moreno et al. demonstrated decreased levels of GSH in patients with ALC. This reduction together with a significant increase in the concentration of oxidized glutathione (GSSG) was the most prominent in Child–Pugh A patients, suggesting a crucial role of oxidative stress in the early stages of ALD. However, researchers from the above-mentioned group noticed increased content of malondialdehyde (MDA) in all examined ALC patients, proportionally to the progression of the disease [10]. Iron overload is a well-known factor participating in chronic alcohol consumption. Former studies conducted on animals proved a synergy between alcohol and iron in promoting lipid peroxidation, which is reflected by an increase in MDA. Moreover, ALC patients were found to present antibodies against CYP2E1 and oxidized phospholipids. According to data already collected in this field, elevation in IgG targeting lipid peroxidation-derived antigens corresponds to TNF-α release and the progression of liver inflammation. Oxidation of lipids is a source of toxic products, e.g., MDA and 4-hydroxy-2-nonenal (4-HNE), which might inhibit the function of numerous mitochondrial proteins, like aldehyde dehydrogenase-2 (ALDH2)—participating in the metabolism of reactive acetaldehyde and 4-HNE, the sirtuin gene family (SIRT), and NAD+-dependent deacetylase, through adduct formation with many amino acid residues [85,86]. The abovementioned lipid peroxides may even alter the cell membrane functions and promote fibrosis due to the activation of HSc, recruitment of cytokines together with neutrophils, and further stimulation of macrophage Kupffer’s cells [87]. Figure 2 shows a complex background of oxidative and nitrosative stress in the course of ALD. 

## 3. Antioxidants, ALD Exacerbation and Signaling Pathways

### 3.1. Acute-on-Chronic Liver Failure and Oxidative Stress

On the other hand, an increased release of reactive oxygen radicals stimulates a defensive pathway, promoting the transcription of antioxidant genes, e.g., peroxiredoxins, sulifredoxin, superoxide dismutases, and glutathione reductase. A key point of this phenomenon constitutes an activation of nuclear translocation of Nrf2 by degrading the cytoplasmic Keap1-Nrf2. Finally, Nrf2 binds to antioxidant response elements and enhances antioxidant defense mechanisms [88,89,90]. Nrf2 knockout mice present destruction of hepatocytes and increased mortality after binge ethanol exposure [91]. ALD is followed by impaired β-oxidation due to excessive oxidative stress. 5-AMP-protein kinase (AMPK) constitutes a key regulator of β-oxidation. Inhibition of AMPK activity by reactive aldehydes (e.g., 4-HNE) contributes to increased steatosis in ALD [92,93,94,95]. In a murine model of ALD, AMPK is covalently modified by reactive aldehydes, reducing its activity. Oxidative stress also regulates AMPK activity. Cells treated with hydrogen peroxide were found to present decreased cellular ATP concentrations and further activation of AMPK. Consequently, phosphorylation and activation of AMPK regulates cellular energy due to increased oxidative stress via β-oxidation in hepatocytes [96,97,98,99,100,101]. Oxidative stress is involved not only in the development of ALD but also in complications in the chronic phase of the disease. Acute-on-chronic liver failure (ACLF) is described as a sudden and acute decompensation of LC, presenting with multiorgan failure and extremely poor survival (28-day mortality rate of 30–40%). It usually occurs in alcoholic- and untreated hepatitis B associated-cirrhosis; bacterial infections and active alcoholism are major causative factors, however, in 40% of cases no triggering event can be identified. ACLF is the manifestation of systemic inflammatory response, acting through diverse mechanisms, e.g., excessive oxidative stress to pathogen- or danger/damage-associated molecular patterns (DAMPs) and/or alteration of tissue homeostasis to inflammation caused either by the pathogen itself or through a dysfunction of tissue tolerance [102,103,104]. The release of bacterial pathogen associated molecular patterns (PAMPs) is the common background of an inflammatory pathway in ACLF, but increased oxidative stress is another unquestionable triggering factor (sterile inflammation) [105,106,107]. Figure 3 shows the overall impact of alcohol intake on the immune system and different susceptible cell subsets. 

### 3.2. Sirtuin-Related Pathways in ALD Natural History

The sirtuin gene family (SIRT) is hypothesized to regulate the aging process and play a role in cellular repair. Sirtuin 6 (SIRT6), NAD-dependent histone deacetylase, has been involved in the course of oxidative stress, also acting as the regulator of longevity, genome stability, metabolism, and inflammation. From a metabolic point of view, SIRT6 suppresses the biosynthesis of triglycerides and cholesterol. Sirt6 systemic knockout results in severe hypoglycemia and premature death [108,109,110,111]. Hepatocyte-specific SIRT6 knockout mice were found to develop hepatic steatosis even on a regular chow diet. Moreover, SIRT6 also serves as a key regulator of inflammation by suppressing pro-inflammatory cytokines (IL-1β, IL-6, and TNF α). SIRT6 has also been proved to alleviate oxidative stress concerning brain ischemia, non-alcoholic fatty liver, and mesenchymal stem cells by regulation of Nrf2 [112,113,114,115]. Nevertheless, little of the data concerns the role of SIRT6 in ALD. Kim et al. tried to evaluate its function in the course of ALD [116]. They found decreased expression of SIRT6 in the livers of ALC patients and ALD mice [117]. The abovementioned researchers additionally created two SIRT6 knockout mouse models and proved that animals with hepatic SIRT6 deficiency are more prone to develop ALD. Interestingly, induction of metallothionein 1 and 2 (MT1 and MT2), anti-oxidative stress genes, by ethanol was significantly impaired in the liver of SIRT6 knockout mice. On the other hand, hepatic SIRT6 overexpression reversed the ethanol induced damage in examined mice. This protection against ALD might be explained by the enhancement of the transcriptional induction of MT1 and MT2 genes by coactivating metal regulatory transcription factor 1 (MTF1). MT1 overexpression decreased hepatic hydrogen peroxide and increased GSH levels among investigated mice. SIRT6 appears to be a promising therapeutic target for oxidative stress in ALD patients [118]. Ethanol consumption is also known to decrease both sirtuin 1 (SIRT1) activity and expression, promoting lipogenesis with inflammation [119,120,121]. Studies based on the improvement of the adenosine monophosphate-activated protein kinase (AMPK)/(SIRT1) pathway in vivo and in vitro concerning alcohol induced hepatotoxicity, revealed the upregulation of SOD and GSH activity and decreased MDA activity [122,123,124,125]. Lee et al. in their recent survey confirmed a crucial role of the SIRT1 pathway in alcohol exposure. They showed that melatonin reduces oxidative stress in ALD due to the induction of SIRT1 expression. Melatonin was found to restore SIRT1 activity in alcohol fed SIRT1-silenced mice [126].

### 3.3. Micro-RNA and Oxidative Stress

The levels of hepatic microRNA (miRNA) might be affected by chronic alcohol consumption and miRNAs interfere with alcohol—induced oxidative stress, liver injury, inflammation, and the development of cancer [127,128,129]. Ethanol upregulates miR-214 and indirectly suppresses cytochrome P450 oxidoreductase and glutathione reductase expression by targeting the 3′-UTR of CYP2E1 transcript [130,131]. ALD in alcoholics with recent excessive drinking is accompanied by downregulated concentration of miR-223, the most common miRNA within neutrophils. On the other hand, genetic deletion of the miR-223 gene enhances ethanol-induced hepatic injury, neutrophil infiltration, ROS generation, and promotes hepatic expression of IL-6 and phagocytic oxidase [132,133,134]. The hepatic accumulation of lipopolysaccharide (LPS) is a natural phenomenon in the course of ALD [135,136,137]. This bacterial antigen activates Toll Like Receptor 4 (TLR4), promoting the transcription of Nuclear Factor kappa B (NF-κB) and the expression of miR-155 and miR-181b-3p. The overexpression of miR-155 and miR-181b-3p causes the release of TNF together with ROS among Kupffer’s cells and hepatic stellate cells [138,139,140,141]. Moreover, miR-291b suppresses Toll interacting protein (Tollip) in Kupffer’s cells, enhancing the TLR4/NF-κB pathway [142]. MiR-155 also regulates lipid metabolism, inhibiting PPAR [143]. Under such circumstances, the overproduction of certain proteins involved in lipid metabolism and uptake occurs (e.g., fatty acid binding protein 4 (FABP4), acetyl-CoA-carboxylase 1 (ACC1) and low-density lipoprotein receptor (LDLR)), triggering the redox state. Of note, miR-34a and miR-217 were proved to target SIRT1 mRNA and inhibit its protein coding in ALD [144,145,146,147].

## 4. Hepatocyte, PRMT1 and Oxidative Stress

Recent surveys highlighted the potentially important role of protein arginine methyltransferase 1 (PRMT1) in oxidative stress related to ALD. Protein arginine methylation belongs to post-translational modifications involved in various pathways, e.g., cell cycle control, innate immune responses, RNA processing, apoptosis, cancer development, and oxidative stress. About 85% of the whole cellular arginine methylation occurs in the presence of PRMT1. The process of methylation is catalyzed by the use of S-adenosyl methionine (SAM) as a methyl donor; methylation involves histone and non-histone proteins. SAM binds and inactivates the catalytic activity of CYP2E1 [148], lowering alcohol-dependent production of superoxide in mitochondria. SAM also increases the synthesis and availability of glutathione, maintaining the mitochondrial respiration rate and mtDNA integrity. Although greater concentrations of SAM have been noted in the serum of ALD patients compared to healthy subjects, a reduced level of hepatic SAM was observed in patients with AH, indicating that an acute inflammatory state leads to hepatic depletion of SAM. A characteristic pattern of ALD concerns reduced ratio of SAM to S-adenosylhomocysteine (SAH)—directly associated with higher intracellular SAH levels. SAH constitutes the product of methionine in the hepatic transmethylation pathway whereby methyl groups from SAM are transferred to a vast number of molecules (e.g., DNA, RNA, biogenic amines, phospholipids) via specific methyltransferases. SAH is a potent competitive inhibitor of most methyltransferases. Abnormal hepatic methionine metabolism is an acquired metabolic abnormality in ALD. Of note, strategies designed to prevent SAH elevation, e.g., betaine administration to ethanol-fed animals, also prevent alcohol-induced lipid droplet accumulation within the liver. Refs. [149,150,151] Arumugam et al. revealed in their study that increased intracellular SAH is sufficient to promote fat accumulation in hepatocytes, which resembles that seen after alcohol exposure [152]. Moreover, PRMT1 serves the role of transcriptional coactivator, participating in splicing and upstream of signal transduction [153,154,155,156,157,158,159,160]. Under physiological conditions, PRMT1 inhibits proliferation of hepatocytes [161]. In alcohol fed mice PRMT1 loses its function and begins to prevent the development of oxidative stress and to promote hepatocyte survival. Interestingly, PRMT1 knockout in alcohol fed mice leads to elevation of ALT, a significant increase in hepatocyte inflammation, death, and liver fibrosis. Zhao et al. determined in their recent study that alcohol is a factor promoting PRMT1 dephosphorylation at S297, resulting in reduced protein methylation in livers of PRMT1 in alcohol fed mice. Phosphorylation at S297 is responsible for PRMT1 target specificity (e.g., expression of TNFα or TRAIL, production of asymmetric di-methyl arginine). However, the expression of oxidative stress response genes due to PRMT1 was found to be phosphorylation independent. Thus, according to the aforementioned study, in terms of exposure to alcohol, PRMT1 directly binds to promoters of these genes, enhancing a recruitment of p300 acetyltransferase to SOD1 and SOD2 promoters and preventing oxidative stress mediated death of hepatocytes. PRMT1 knockout in alcohol fed mice was followed by 40–60% reduction of the oxidative stress response genes [162]. Therefore, PRMT1 plays the role of adaptive factor in the course of ALD.

## 5. Oxidative Stress and Epigenetic Background

Direct modification of DNA is not the only possible direction of change in gene expression due to alcohol intake. Lifestyle, combined with environmental factors, might be involved in genetic modification in the course of redox state, as well. Epigenetic regulation concerns DNA and histone protein modifications and changes caused by non-coding mRNAs. DNA methylation belongs to the most common epigenetic changes, directly influencing the expression of a gene [163,164]. Furthermore, histone proteins might undergo acetylation and deacetylation via enzymatic modification caused by histone deacetylases (HDACs) and histone acetyl transferase (HAT). As a result, the structure of chromatin becomes unfolded or compacted. The redox state can inhibit the expression of HDACs by PI3Kδ, a signaling molecule involved in various inflammatory signaling pathways. The inhibition of PI3Kδ in patients with AH was even found to increase the response on steroids in this group [165,166]. In addition, alcohol intake increases gene-selective acetylation of histone H3 at lysine 9 (H3K9), levels of enzymes mediating histone acetylation, and results in a generalized increase in DNA methylation [167,168]. These epigenetic-derived effects of ethanol consumption directly modify inflammatory response, through crucial pro-inflammatory cytokines, such as TNF-α, which is silenced by H3K9 methylation and activated by H3K9 acetylation [169,170].

## 6. Redox State in the Liver and Potential Pharmacological Strategies

The complex role of ROS in the course of ALD gives a potential chance to create a targeted anti-inflammatory pharmacological strategy for its treatment. The attempts to reduce oxidative stress within mitochondria concern the use of SAM. Especially short-term treatment of the acute mitochondrial stress observed in AH can be considered as a potential indication for the use of SAM. A combination of SAM and prednisolone for the treatment of severe AH showed improved response rate assessed by Lille score and a reduction in hepatorenal syndrome [171]. However, no statistically significant difference in 28-d mortality was noted. Similarly, long term SAM treatment in patients with ALD does not appear to be clinically effective, with no change in overall mortality [172,173]. Uncoupling proteins (UCPs) are known to be strongly associated with mitochondrial stress in ALD. Overexpression of UCP2 reduced apoptosis and oxidative stress in vitro, however this issue requires further studies [174]. N-acetylcysteine (NAC), an antioxidant therapy that derives cysteine for glutathione synthesis, was tested in patients with AH. Initial trials did not show a significant survival benefit [175,176]. Nevertheless, a recent study on the combination of NAC and prednisolone, presented a reduction in infective complications and 1-month mortality [177]. Additionally, antioxidant therapy including zinc and other trace elements turned out to be clinically beneficial in patients with AH [178]. However, interpretation was hampered by the use of a variety of antioxidants at differing concentrations and durations [179]. Epigenetic background can be also perceived as a target for the treatment of redox state. Studies concerning HDACs have not yet been done on patients with ALD but in vitro results indicate an antioxidant effect of HDAC inhibition with upregulation of Nrf2 expression [180].

## 7. Conclusions

The redox state remains a crucial pathology involved in the pathological appearance of ALC. Nowadays we are able to capture the early beginning of this harmful cascade in the liver. This opens up new possibilities of treatment, which might revolutionize the management of patients with this type of liver disorder. The future of management of the redox state seems to stay before novel therapies, focused on miRNA signaling, epigenetics, and signaling pathways. It appears that we are able to capture oxidative stress at an earlier and earlier phase. For this reason, the attitude toward ALC patients should be more involved, based on up-to-date knowledge from a molecular point of view and on epigenetic features, highlighting the individual profile of the disease in each person.

## Figures and Tables

**Figure 1 jcm-10-03011-f001:**
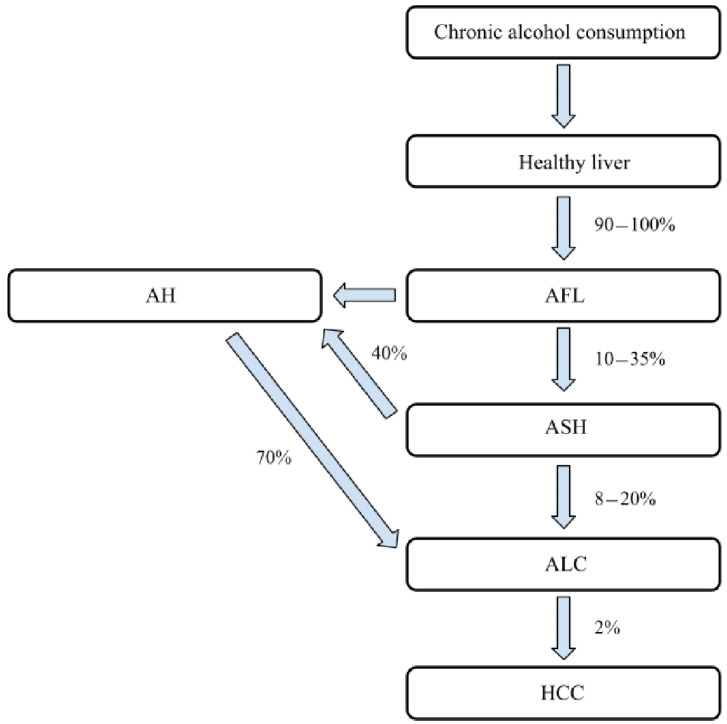
Natural history of alcohol-related liver disease. (AFL—alcoholic fatty liver, AH—alcoholic hepatitis, ASH—alcoholic steatohepatitis, ALC—alcohol-related liver cirrhosis, HCC—hepatocellular carcinoma). Chronic alcohol consumption affects healthy liver, leading to the development of AFL in 90–100% of people; 10–35% AFL patients progress to ASH and ALC is the complication in 8–20% of them. Finally, 2% of cirrhotic patients develop HCC. AH is an additional stage of ALD, which might develop from AFL or ASH and directly progresses into ALC (in up to 70% cases).

**Figure 2 jcm-10-03011-f002:**
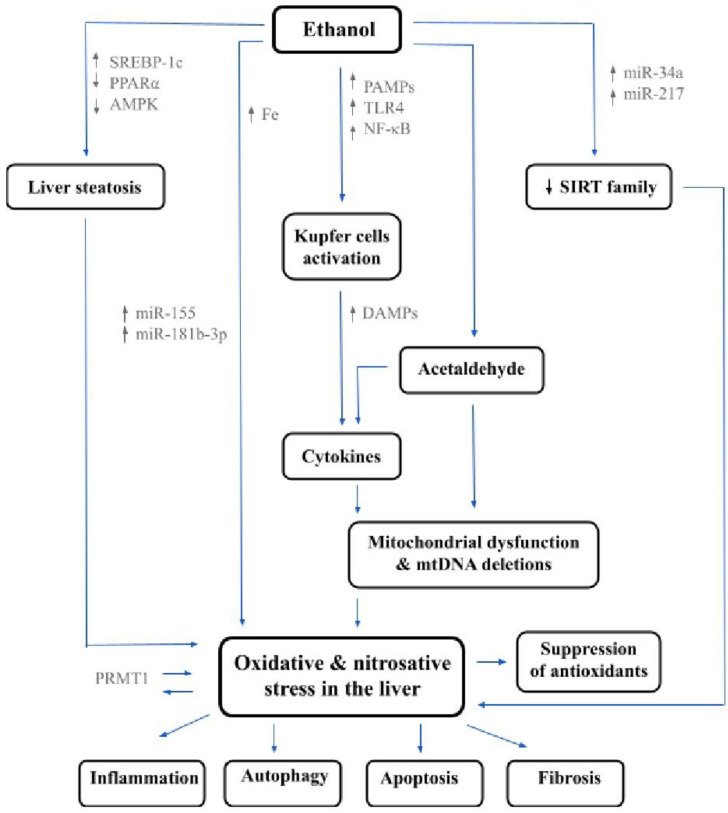
Oxidative and nitrosative stress in the course of alcohol-related liver disease—main pathways. (SREBP-1c—sterol regulatory element-binding protein 1, PPARα—peroxisome proliferator-activated receptor α, AMPK—AMP-activated protein kinase, Fe—iron, PAMPs—pathogen associated molecular patterns, TLR4—toll-like receptor 4, NF-κB—nuclear factor κ B, DAMPs—danger/damage-associated molecular patterns, SIRT—sirtuin gene family, PRMT1—protein arginine methyltransferase 1, miR—microRNA, mtDNA—mitochondrial DNA).

**Figure 3 jcm-10-03011-f003:**
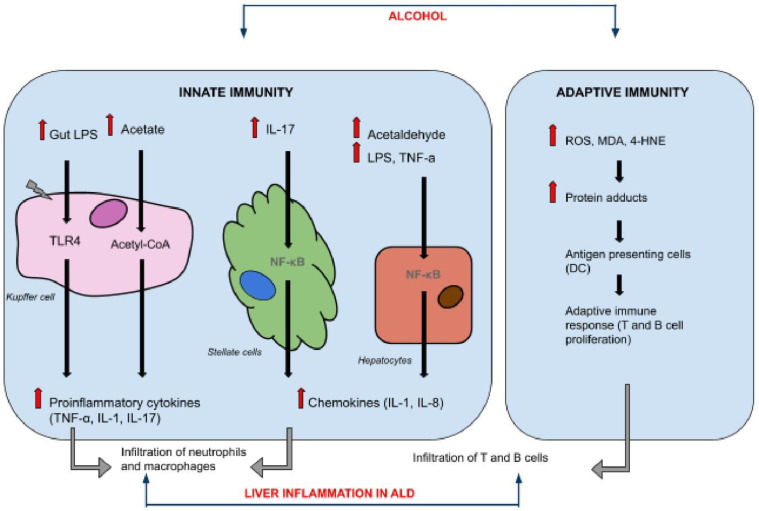
Alcohol, immune system and crucial cells involved in signaling pathways. Alcohol intake directly influences innate and adaptive immune behaviors, being responsible for the disturbed course of various physiological processes within different type of cells. LPS—lipopolysaccharide, TLR-4—IL—interleukin, TNF-α—tumor necrosis factor alpha, NF-κB—nuclear factor κ B, ROS—reactive oxygen species, MDA—malondialdehyde, 4-HNE—4-hydroxy-2-nonenal—malondialdehyde.

**Table 1 jcm-10-03011-t001:** Molecules involved in the development of redox state in case of ALD.

Protein/Gene/Molecule	Role in ALD—Induced Oxidative Stress
miR-212 and iNOS	alcohol-induced gut leakiness
SREBP-1c and PPARα	promotion of liver steatosis
reactive aldehydes (e.g., 4-HNE)	promotion of liver steatosis
PAMPs and DAMPs	progression of inflammation
SIRT family	progression of oxidation and inflammation
miR-214	suppression of cytochrome P450
miR-223	involved in neutrophils infiltration and ROS generation
miR-155 and miR-181b-3p	LPS-mediated inflammation
miR-291b	involved in TLR4/NF-κB pathway
miR-34a and miR-217	inhibits the expression of SIRT1

miR—micro-RNA, iNOS—inducible nitric oxide synthase, SREBP-1c—sterol 191 regulatory element-binding protein 1, 4-HNE—4-hydroxy-2-nonenal, PAMPs—pathogen associated molecular patterns, DAMPs—danger/damage-associated molecular patterns, SIRT—sirtuin gene family, LPS—lipopolysaccharide, TLR4—toll-like receptor 4, NF-κB—nuclear factor κ B, SIRT—sirtuin gene family.

## Data Availability

Data is contained within the article.

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
