# Peer review of "Oxidative Stress—A Key Player in the Course of Alcohol-Related Liver Disease"

_jcm, 2021, doi:10.3390/jcm10143011_

Round 1
Reviewer 1 Report
The review addresses underlying mechanisms associated with redox state in the course of alcohol-related liver disease. As the topic is known and widely discussed, each new study should bring new value to the knowledge and understanding of the issue. Unfortunately, despite the efforts of the authors and the attempts to present the latest data, the manuscript has several flaws, as described below.
So far, several reviews regarding oxidative stress in alcohol-related liver injury have been released, including one with strikingly similar title: Tan HK, Yates E, Lilly K, Dhanda AD. Oxidative stress in alcohol-related liver disease. World J Hepatol 2020; 12(7): 332-349. Surprisingly, the authors did not refer to it. That study mentioned above contains a description of the selection of literature and fine graphical presentation (in contrast to the present study submitted for review ).
The first question that arises is how this review study is different from the previous ones? What research will be discussed? In vitro, preclinical, clinical research? Is it an update of previous review papers?
The manuscript is quite chaotic. As an example - the sentence: „iNOS can be even involved in alcohol-induced gut leakiness; it was found to be the reason for miR-212 overexpression and further intestinal hyperpermeability [57].” (page 3, line 110) – What genes and associated pathways is this particular miR involved in?
Clinical data should have been included in the tables. Without such summaries, it isn't easy to follow the text. The authors mix preclinical and clinical data, patients with animals. Moreover, enzymes and genes, epigenetic modifications, biomarkers from patients' blood, and therapies with antioxidants should be presented.
The graphics in the manuscript are of poor quality. Figure 1 is not sufficiently well described in the legend. What are these percentages? Does alcohol lead to a healthy liver? Do we need figure 2 as a separate diagram? Overall, figures should be more original and contain new information. It is worth adding a figure presenting intracellular compartments and signaling pathways involved in alcohol oxidative stress within hepatocytes.
It is worth adding the table of genes and miRNAs discussed in the manuscript.
It would be worth distinguishing changes in the redox state occurring in the first stage and end-stage of alcoholism.
I cannot consent to the authors' interpretation of the PRMT-1 role in alcohol-related liver injury. One of the main challenges linked with methionine metabolism in the course of this disorder is S-adenosylmethionine deficiency and the buildup of S-adenosylhomocysteine. So why didn't the authors discuss this issue? Instead of this, they are focused on the one enzyme, protein arginine methyltransferase, and in addition, they refer to papers describing the significance of PRMT-1 in neoplastic transformation (hepatocellular carcinoma). However, they do not explain this and imply generalized conclusions for the course of ALD.
The authors did not present any new conclusion that could be drawn from the presented data. What guidelines for clinicians can they suggest?
Second author contribution (Tomasz Lach) is not listed.
What is this non-published data file? It is poorly formatted (fonts cannot be seen).
The manuscript text should be read and revised by an English native speaker.
Author Response
- A paper by Tan et al. (Tan HK, Yates E, Lilly K, Dhanda AD. Oxidative stress in alcohol-related liver disease. World J Hepatol. 2020 Jul 27;12(7):332-349. doi: 10.4254/wjh.v12.i7.332. PMID: 32821333; PMCID: PMC7407918.) was included in the references section (position no. 175).
- A presented review constitutes an update of previous publications of this type, referring to the majority of available papers devoted to the topic of redox state in patients with alcohol-related liver disease. Despite a growing number of similar populations, it appeared to us that there is still a great demand for comprehensive publications in this field. Our aim was to discuss various types of investigations to include into the paper possibly most up-to-date information.
- The part discussing the role of miRNA-122 and iNOS was improved. We tried to present in the paper different types of mechanisms involved in the pathological appearance of ALD-derived oxidative stress.
- Our paper has the character of a review not a meta-analysis. Clinical data extracted from various studies are presented just to broaden the idea of redox state and ALD. Those studies do not come with sufficient amount clinical data to present them in a graphic way.
- Sections presenting epigenetic modifications and antioxidants in ALD were created (no. of sections: 5 and 6, respectively).
- Figure 1 was removed and we created a new one, which contains different pathways that take place in various cells of ALD patients.
- We added a table containing discussed molecules/factors important for the course of ALD in the paper.
- The data concerning a differentiation between the first and last stage of alcoholism were added.
- Details concerning S-adenosylmethionine were placed in the paper. Moreover, we referred to S-adenosylhomocysteine in the course of ALD. Looking for novel information concerning ALD, PRMT-1 paid our attention because of relatively limited data on its certain role in ALD population. and it appears quite rarely in similar reviews. Therefore, it has appeared to us important to include it in a current article.
- We modified the section of conclusions, highlighting a need of understanding a complex character of redox state and it's potential implication in the future management of these patients
- Contributions of Tomasz Lach have been listed.
- The manuscript was revised by a native speaker and linguistic errors have been corrected.
All corrections are marked in color.
Reviewer 2 Report
- The authors should include oxidative stress related epigenetic changes in ALD. This should include alterations in DNA methylation patterns and global histone acetylation due to ROS in ALD.
- The authors should also include a separate section on the signaling pathways implicated in ROS induced ALD.
- The readers would benefit if there was a section on antioxidant therapies.
Author Response
- Epigenetic changes in the course of ALD with an impact on DNA methylation and histone acetylation were added.
- The title of the third section of the paper was modified (Antioxidants, ALD exacerbation and signalling pathways), because this part already contains data devoted to signaling pathways.
- A section of antioxidant therapies was added.
All corrections are marked in color.
Round 2
Reviewer 1 Report
The paper has improved substantially, and most of my questions and concerns have been addressed. However, I still believe that this manuscript could be better structured and more focused on selected signaling pathways. There are a few points the authors might look into before this manuscript is ready for publication.
Expand on the abbreviation ZO-1 (zonula occludens-1);
There are many sirtuins with broad functional diversity; please clarify in Table 1 that „miR-34a and miR-217” are involved in the SIRT-1-mediated signaling pathway;
Please provide Figure 3 with higher resolution, fonts are not clear;
Consider reducing the list of references. Are 185 items necessary? If some references are repeated in other reviews you cite, you can limit the number of source papers.
Author Response
- In the first round of review we tried to do our best during the modification of a structure of the manuscript and analyzing its content with a special attention to signaling pathways.
- The abbreviation of ZO-1 was expanded.
- The role of miR-34a and miR-217 in the course of ALD was clarified.
- The Figure 3 was resized.
- The list of references was reduced. Positions: 135, 136, 156 and 160 were removed.
All introduced changes were written with blue colour.
Reviewer 2 Report
The authors have addressed the reviewer’s concerns.
Author Response
Thank you for the positive feedback.